# Molecular Roles of NADPH Oxidase-Mediated Oxidative Stress in Alzheimer’s Disease: Isoform-Specific Contributions

**DOI:** 10.3390/ijms252212299

**Published:** 2024-11-15

**Authors:** Junhyung Kim, Jong-Seok Moon

**Affiliations:** 1Department of Integrated Biomedical Science, Soonchunhyang Institute of Medi-Bio Science (SIMS), Soonchunhyang University, Cheonan 31151, Chungcheongnam-do, Republic of Korea; hotdog58@naver.com; 2Department of Pathology, College of Medicine, Soonchunhyang University, Cheonan 31151, Chungcheongnam-do, Republic of Korea

**Keywords:** NADPH oxidases, Alzheimer’s disease, oxidative stress, reactive oxygen species

## Abstract

Oxidative stress is linked to the pathogenesis of Alzheimer’s disease (AD), a neurodegenerative disorder marked by memory impairment and cognitive decline. AD is characterized by the accumulation of amyloid-beta (Aβ) plaques and the formation of neurofibrillary tangles (NFTs) of hyperphosphorylated tau. AD is associated with an imbalance in redox states and excessive reactive oxygen species (ROS). Recent studies report that NADPH oxidase (NOX) enzymes are significant contributors to ROS generation in neurodegenerative diseases, including AD. NOX-derived ROS aggravates oxidative stress and neuroinflammation during AD. In this review, we provide the potential role of all NOX isoforms in AD pathogenesis and their respective structural involvement in AD progression, highlighting NOX enzymes as a strategic therapeutic target. A comprehensive understanding of NOX isoforms and their inhibitors could provide valuable insights into AD pathology and aid in the development of targeted treatments for AD.

## 1. Introduction

Alzheimer’s disease (AD) is known as one of the representative neurodegenerative diseases [1]. As the predominant type of dementia, AD is characterized by memory impairment, cognitive decline, and behavioral deficits [2]. Two distinctive neuropathological features of AD are the extracellular accumulations known as the deposition of amyloid-beta (Aβ) plaques and intracellular structures called neurofibrillary tangles (NFTs), composing hyperphosphorylated tau protein [3]. For decades, extensive research efforts have focused on the pathophysiology of AD, with many studies aimed at uncovering its underlying mechanisms and risk factors. In recent years, numerous studies have indicated that oxidative stress is a significant factor in both the onset and developments of AD [4]. Oxidative stress, which increases with brain aging, arises from an imbalance in the redox state due to an excessive production of reactive oxygen species (ROS) or dysfunction of the antioxidant system [5,6]. This stress has occurred when free radicals exceed the antioxidant capacity of the central nervous system (CNS), and this involves the disruption of calcium homeostasis [7]. In summary, the excessive production of ROS leads to oxidative stress, which plays a critical role in damaging cellular components during AD pathogenesis. Despite the emerging role of oxidative stress in the pathogenesis of AD, the mechanisms by which redox balance is altered and free radicals are generated are still unclear.

Several potential enzymes contribute to the production of ROS, including nicotinamide adenine dinucleotide phosphate (NADPH) oxidase, cyclooxygenase, xanthine oxidase, microsomal enzymes, lipoxygenases, and nitric oxide synthase (NOS) [7,8]. Among these ROS-related factors, NADPH oxidase (NOX) exhibits remarkable and rapid responsiveness to increase ROS levels through the stimulation of various growth factors, including cytokines like platelet-derived growth factor (PDGF) and nerve growth factor (NGF) [9].

NOX enzymes are a family of important enzymes that produce ROS. In most mammals, especially in humans, the NOX family of enzymes comprises seven isoforms named NOX1, NOX2, NOX3, NOX4, NOX5, dual oxidases (DUOX) 1, and DUOX2 [10]. All NOX family enzymes are transmembrane proteins that traverse the membrane six times and produce O2^•−^ from oxygen through NADPH oxidation, utilizing a heme-dependent mechanism [11]. In general conditions, the phagocytic NOX catalyzes the generation of superoxide (O2^•−^), which aids to eliminate invading microbes during phagocytosis, to support host defense and to contribute to redox signaling [12]. In activation conditions, O2^•−^ generates H_2_O_2_ and other ROS that contribute to the destruction of pathogens [10]. However, the excessive production of free radicals, including hydrogen peroxide and peroxynitrite, a reactive oxidant produced from nitric oxide (NO) and superoxide, can exacerbate the progression of the physiological and pathological conditions in AD [13]. Additionally, microglial NOX is a primary source of ROS and a key contributor to the extensive oxidative damage observed in both human AD brains and AD mouse models [6,14]. Given these various reasons, recent studies have emphasized the potential significance of NOX isoforms in neurodegenerative disorders, including AD and Parkinson’s disease (PD) [15].

Understanding the roles of AD-related NOX isoforms could provide valuable insights into the relationships underlying the pathology of AD, particularly regarding oxidative stress and neuroinflammation. In this review, we explore the characteristics of NOX isoforms, their structural features relevant to AD, and their inhibitors. By examining current evidence, we aim to unravel the potential relevance of respective NOX isoforms in AD pathogenesis, potentially paving the way for the development of targeted therapeutic strategies.

## 2. Structure of NOX Isoforms

The seven members of the NOX/DUOX isoforms share a conserved structure, featuring a catalytic core with six transmembrane α-helices that anchor two heme units via histidine residues, complemented by cytosolic domains with binding sites for FAD and NADPH, which collectively facilitate electron transfer and subsequent ROS generation [16].

NOX1, NOX2, and NOX3 interact with the small transmembrane protein p22phox, also known as the human neutrophil cytochrome b light chain (CYBA) [17,18]. The activation of NOX1 and NOX3 requires interaction with cytosolic subunits NADPH oxidase activator 1 (NOXA1) and NADPH oxidase organizer 1 (NOXO1), which are homologous to p67phox and p47phox in NOX2, respectively [19,20]. Structurally, the constitutive activation of NOX1 due to NOXO1′s membrane localization and its dependence on NOXA1 for activation contributes to chronic oxidative stress and Aβ-induced neurotoxicity in AD brains [21]. NOX2, the most studied isoform in AD, is implicated in microglial activation, neuroinflammation, and oxidative damage in AD brains. The structural involvements of the NOX2 isoform in AD pathology have also been elucidated through postmortem analyses of brain tissues and will be further discussed in the following sections.

While NOX4 interacts with p22phox, it differs from other NOX enzymes as it functions without requiring cytosolic subunits for activation [22]. This structural autonomy of NOX4, combined with its ability to generate H_2_O_2_, allows for continuous low-level ROS production, making it a unique contributor to chronic oxidative stress in AD pathology [23,24]. Due to its structural distinctiveness, recent studies have increasingly focused on NOX4 in AD pathogenesis, highlighting its roles in tau hyperphosphorylation and cognitive decline in AD models.

NOX5 does not engage with p22phox, instead forming homo- or multimeric structures [25]. Also, NOX5, DUOX1, and DUOX2 exhibit additional structural features including EF-calcium-binding domains and the peroxidase domains present in DUOX enzymes [26,27]. Unlike other NOX isoforms, NOX5, DUOX1, and DUOX2 do not require assistance from other proteins for activation [28]. They are activated by the binding of Ca^2+^ ions to their cytosolic EF-calcium-binding domains [29]. Specifically, NOX5 possesses an N-terminal extension with four Ca^2+^-binding EF hands, while DUOX1 and DUOX2 have two N-terminal EF hands, an extra N-terminal transmembrane domain, and a peroxidase homology region at their N-terminus [27,28,29]. The Ca^2+^-binding structure of NOX5, which is essential for its calcium-dependent activation, has been shown to impact blood–brain barrier integrity and memory loss in aging mice, indicating a potential role in AD progression [30,31]. The unique structural feature of an N-terminal peroxidase-like domain in DUOX1 and DUOX2 enables direct hydrogen peroxide generation, potentially contributing to the altered redox balance observed in brains of AD [32,33,34]. Also, DUOX1 and DUOX2 require interaction with dual oxidase maturation factor (DUOXA) 1 and DUOXA2, respectively, for proper maturation and function [35].

In summary, the distinct structural characteristics of each NADPH oxidase isoform (Figure 1 and Table 1) underscore their unique coIntributions to oxidative stress, neuroinflammation, and neurodegeneration in AD. While NOX2 and NOX4 are currently the focus of most AD-related NOX research, emerging evidence suggests that other isoforms may also play a role in AD pathogenesis. Further investigation into the specific functions of each NOX isoform in AD could provide the foundation for understanding how each isoform plays specific roles in AD progression.

## 3. The Roles of NOX Isoforms in AD

Previous studies have shown the elevation of NOX activity in the cortex of patients with mild cognitive impairment (MCI), suggesting a potential significance of NOX in the pathogenesis of AD [36]. Among the NOX isoforms, NOX2 and NOX4 are the predominant isoforms implicated in AD pathology based on their elevated expression and activity in brain regions [37]. However, in addition to NOX2 and NOX4, experimental evidence regarding the roles of other isoforms such as NOX1, NOX3, NOX5, DUOX1, and DUOX2 will also be explored. While these isoforms have been less extensively studied in the context of AD, emerging evidence suggests that they may have some degree of association with the disease process. In this section, we will review the potential roles and contributions of all NOX isoforms in AD pathology by evaluating current evidence and their potential implications in the progression of AD. By examining the full spectrum of NOX isoforms, we aim to provide a comprehensive understanding of how these enzymes might contribute to oxidative stress, neuroinflammation, and neurodegeneration in AD (Figure 2 and Table 2).

### 3.1. The Roles of NOX1 in AD

NOX1 produces superoxide and shares significant structural and functional similarities with NOX2 and NOX3, particularly in terms of their homologous regulatory subunits and regulation by Rac GTPase [38,39,40]. Previous studies have shown that NOX1 is highly expressed in gastrointestinal epithelial cells [39,40]. Recent studies have reported the potential role of NOX1 in AD pathogenesis, particularly through microglial activation and neuroinflammation [41]. In this study, unlike the phagocytic NOX2 system, NOXO1 and NOXA1 are constitutively associated with NOX1 at the membrane, facilitating more rapid and sustained ROS production in AD-affected tissues [41]. This structural arrangement, combined with the activation of NOX1 by small GTPases like Rac1, suggests a mechanism for chronic oxidative stress and heightened ROS generation in response to cellular stressors, which may contribute to the progression of AD [41]. In studies using LPS-induced mouse models, NOX1 in microglia contributes to synaptic damage through the production of inflammatory mediators and oxidative stress, paralleling processes observed in AD [42]. Another study showed that mRNA levels of NOX1 are increased in the brain at the early stages of AD patients [42]. This upregulation of NOX1 in AD has been identified as a major contributor to increased oxidative injury, potentially leading to mitochondrial dysfunction and subsequent energy failure in neurons [43]. In preliminary studies, rat cerebellar granule neurons exposed to transient oxidative stress (induced by 48 h of exposure to 50 mM ethanol) showed elevated NOX1 expression, suggesting that oxidative injury can drive NOX1 upregulation, further linking it to the early stages of AD pathogenesis [43]. These findings suggest the possibility that the roles of NOX1 can contribute to the increase in oxidative stress and injury that reduces the function of Complexes IV and V in the mitochondrial electron transport chain during AD [43].

### 3.2. The Roles of NOX2 in AD

NOX2 was initially identified in phagocytic cells and serves as the prototype for other NOX enzymes [44,45]. NOX2 is the most prominent isoform found in neurons, microglia, and astrocytes [45]. A study reported that elevated NOX2 activity is correlated with increased levels of Aβ and excessive oxidative stress [46]. With structural involvement, post-mortem analyses of AD brains reveal that NOX2 activation, indicated by the translocation of its subunits such as p47phox and p67phox to the cell membrane, significantly contributes to oxidative damage and neuroinflammation [32,37,44]. This process is further supported by findings that Aβ induces NOX2 activation in microglia, leading to the production of ROS and pro-inflammatory cytokines, which exacerbate synaptic loss, neuronal damage, and cognitive decline [24]. For instance, in AD models, NOX2-dependent ROS generation in microglia has been associated with elevated levels of IL-1β, TNF-α, and IL-6 [21]. These cytokines are consistently found to be upregulated in the brains of AD patients and animal models [21]. Specifically, IL-1β levels have been shown to increase in parallel with NOX2 activation in the frontal and temporal cortex during AD progression [21]. Notably, the inhibition of NOX2 can prevent Aβ-induced oxidative stress, glucose hypo-metabolism, and network hyperactivity, underscoring the critical role of NOX2 in AD pathology [47]. Further study has shown that NOX2 activity is negatively correlated with cognitive status in humans, suggesting that increased NOX2 activity is associated with cognitive decline [24]. In animal models, NOX2 has been implicated in vascular dysfunction associated with AD pathology through the observation that the reduced cerebral blood flow in Tg2576 transgenic mice overexpressing Aβ, which was not observed in Tg2576 mice lacking NOX2 [48]. Furthermore, the enhanced activity of NOX2 in AD, driven by Aβ-induced microglial activation and subsequent ATP release, amplifies oxidative stress and neuroinflammation, thereby contributing to AD progression [49]. Therefore, these findings suggest that the roles of NOX2 contribute to brain oxidative stress and neuroinflammation via microglial responses to Aβ stimulation during AD.

### 3.3. The Roles of NOX3 in AD

NOX3, a member of the superoxide-producing NOX family, was first recognized as the third oxidase predominantly found in the fetal human kidney [40]. This oxidase has been recently shown to be highly expressed in the neurons of the inner ear, where it plays a crucial role in otoconia formation and balance perception [50]. However, recent studies have reported that NOX3 is detected in various organs and cell types [5]. Also, mRNA levels of NOX3 are elevated in the frontal cortex of AD brains compared to normal brains, suggesting potential involvement in neurodegeneration [43]. While the expression of NOX3 in the CNS has been less studied compared to other NOX isoforms, its ability to generate superoxide, and its regulatory mechanisms such as p22phox, NOX organizer 1 (NOXO1), and p67phox, which are involved in oxidative stress pathways [50], this unique regulatory mechanism allows NOX3 to generate superoxide even when gp91phox and NOX1 are inactive [50]. Studies have shown that while p47phox and NOXO1 can enhance NOX3 activity, the small GTPase Rac, which is crucial for other NOX isoforms, appears to be dispensable for NOX3 function [43,50]. The elevated levels of NOX3 mRNA in AD brains may indicate its potential role in contributing to oxidative stress and neuroinflammation, thereby influencing the progression of AD [50]. Although the presence of NOX3 in the brain of AD was identified, the roles of NOX3 are not well established yet. Further studies are needed to determine the roles of NOX3 during AD.

### 3.4. The Roles of NOX4 in AD

NOX4 has emerged as a significant NOX isoform in AD pathogenesis through its contribution to oxidative stress and neurodegeneration. Unlike other NOX isoforms, NOX4 is constitutively active and primarily generates hydrogen peroxide, which is rapidly converted from superoxide [51]. In AD brains and relevant animal models, NOX4 expression is significantly elevated, particularly in neurons and astrocytes, correlated with increased levels of Aβ and hyperphosphorylated tau, hallmark features of AD [52,53]. Recent studies have further implicated NOX4 in astrocytic ferroptosis, a regulated form of cell death driven by iron-dependent lipid peroxidation [25]. This is evidenced by the elevated levels of oxidative stress markers such as malondialdehyde (MDA) and 4-hydroxynonenal (4-HNE) in AD brains [54]. Additionally, the levels of NOX4, an upstream molecule of ferroptosis in astrocytes, were increased in the 4-HNE-positive astrocytes in the cerebral cortex of brains of AD patients [25]. The interaction between NOX4-mediated ferroptosis and AD progression suggests a significant pathway for neurodegeneration in AD and provides new insights into potential therapeutic strategies [16]. In neurons, NOX4 knockdown reduced the accumulation of pathological tau and improved the macroautophagic flux. Also, NOX4 knockdown decreased neurotoxicity and prevented cognitive decline, after the induction of tauopathy [52]. These findings suggest that the roles of NOX4 can contribute to the activation of reactive astrocytes, ferroptosis-induced astrocytes degeneration, and neuronal tauopathy during AD.

### 3.5. The Roles of NOX5 in AD

NOX5 was reported in 2001 as the latest identified NOX isoform and most distinct isoform in the NOX isoforms [55,56]. Unlike with NOX1, NOX2, NOX3, and NOX4 enzymes, NOX5 activity does not require the presence of accessory proteins and only depends on the increase in intracellular calcium at their cytosolic EF-calcium-binding domains [57,58]. While NOX5 is primarily expressed in tissues including the spleen and testis, the expression of NOX5 in the brain has been observed in cell types such as oligodendrocytes and glioblastoma (GBM), a type of cancer that starts in astrocytes [59,60].

Recent studies indicate that NOX5 may contribute to AD pathogenesis through neuroinflammation, oxidative stress, blood–brain barrier (BBB) disruption, and cognitive decline [30,61]. Experiments with genetically modified knock-in (KI) mouse model expressing the human NOX5 gene and in vitro tissue cultures demonstrated that both reoxygenation and calcium overload elevated brain ROS levels in a manner dependent on NOX5 [61]. Even though the expression of NOX5 in humanized KI mice did not affect survival, it led to impaired memory and cognitive deficits, while significantly increasing the levels of inflammatory components COX2 and TXA2S [30]. Also, other studies have reported that the abnormal accumulation of NOX5 protein levels leads to elevated ROS and cell death in AD pathology [62]. In terms of the cause of the behavioral changes and memory loss, the increased NOX5 activity has been linked to oxidative damage and BBB disruption, which are associated with the pathological features of AD [30,63]. This evidence suggests the roles of NOX5 that may contribute to memory loss through the alteration of the integrity of the BBB during AD.

### 3.6. The Roles of DUOX1 and 2 in AD

Dual oxidases 1 and 2 (DUOX1 and 2) are unique members of the NOX family, distinguished by their additional structural features including a gp91phox domain, an extracellular peroxidase domain, and EF-calcium-binding motifs [28,37,64]. These two subunits have 83% similarity, which indicates a significant structural similarity between them [26]. These enzymes primarily generate hydrogen peroxide and interact with specific activator proteins (dual oxidase maturation factor (DUOXA) 1 and 2), which are crucial for their targeting and function [32,35]. Both DUOX1 and DUOX2 are commonly examined in the thyroid and respiratory tract epithelium [65,66]. Recently, studies related to DUOX enzymes have been extended to other body systems [32]. DUOX enzymes have been identified in oligodendrocytes and GBM in humans, as well as in rodent brains [67,68]. Additionally, another study reported that DUOX enzymes are linked to ROS-dependent immunity [32,69]. A recent study showed that DUOX contributes to oxidative stress in neurons and influences the lifespan of *Drosophila melanogaster* [33]. Notably, the neuronal knockdown of DUOX reduces oxidative damage and extends lifespan, demonstrating its role as a significant source of ROS [33]. Furthermore, DUOX activity increases in aged brains and in models of Alzheimer’s disease, suggesting a connection between chronic inflammation and oxidative stress in neuronal aging [33]. Additionally, another study showed that in transgenic flies expressing Aβ42 and tau, mRNA levels of DUOX were significantly elevated compared to control flies [70]. Also, increased DUOX expression correlated with age-dependent neurodegeneration in Drosophila AD models, as indicated by high levels of vacuolization observed in brain sections [70]. These findings suggest that the roles of DUOX may contribute to neuronal degeneration during AD; however, further human studies on DUOX 1 and 2 are needed.

## 4. NOX Inhibitors in AD

Numerous studies have reported on the significant involvement of various NOX isoforms in AD pathogenesis. Therefore, understanding the inhibitors of these NOX isoforms could significantly contribute to new therapeutic approaches for AD. However, the complexity of NOX functions and the broad impacts of some inhibitors present both opportunities and challenges for their clinical applications. These inhibitors can be broadly categorized into non-specific and isoform-specific inhibitors, each with their own advantages and limitations. Among the various NOX inhibitors in AD, the earliest discovered and most well-known are diphenyleneiodonium (DPI) and apocynin [71]. DPI suppresses superoxide production by inhibiting flavoproteins, thereby affecting various enzymes beyond NOX, including nitric oxide synthase and xanthine oxidase [72]. While DPI has shown potential in reducing neuroinflammation and oxidative stress in AD models, its broad inhibitory effects on multiple flavin-dependent enzymes limit its clinical potential as a specific NOX inhibitor [73,74,75]. This lack of specificity poses a risk for off-target effects, as the inhibition of other flavoproteins could lead to unexpected side effects in a clinical setting. Therefore, although DPI has been valuable in experimental AD models, its broad activity makes it less suitable for human therapies targeting NOX in AD. Also, apocynin has demonstrated neuroprotective effects in various neurodegenerative diseases by reducing oxidative stress and inflammation through the interference of intracellular translocation in p47phox and p67phox subunits [76]. Unlike DPI, apocynin’s action appears more targeted within the NOX pathway, which may reduce broader systemic effects. Recent studies indicate that apocynin demonstrates efficacy in attenuating the progression of AD by Aβ accumulation, oxidative stress, and neuroinflammation, potentially through the modulation of BACE1 activity and transcription factors such as Nrf2 and NF-kB [77,78,79]. However, the exact mechanism of action of apocynin in AD is still not fully understood, and its efficacy may vary depending on the specific pathological context. Further research is needed to elucidate its long-term effects and optimal dosing strategies in AD patients. In addition to DPI and apocynin, other NOX inhibitors such as GKT137831, VAS2870, and Nox2ds-tat have attracted attention for their potential effects on AD pathology. GKT137831, described as a preferential direct inhibitor of NOX1 and NOX4, has shown excellent tolerability and reduced chronic inflammation markers in clinical trials [80]. This selective inhibition of NOX1 and NOX4 may offer a more targeted approach in managing AD pathology by reducing neuroinflammation while maintaining better tolerability. However, further trials are necessary to establish its efficacy and safety in long-term AD treatment. NOX2ds-tat, an 18-amino acid peptide and the first biological NOX inhibitor, has demonstrated significant efficacy in AD models [81]. When administered to aged Tg2576 mice or in studies using NOX2 knockout mice crossed with Tg2576 mice, NOX2ds-tat markedly reduced oxidative stress, enhanced neurovascular function, and alleviated behavioral impairments associated with the AD progression [50]. These findings suggest that NOX2ds-tat not only has the potential to reduce AD-related oxidative stress but also improves neurovascular health, a critical factor in AD pathology. VAS2870, identified by Vasopharm GmbH as a pan-NOX inhibitor, inhibits all NOX isoforms except NOX3 in cellular assays and particularly shows a 40% to 70% inhibition of NOX2 and NOX4, meaning it is suitable for evaluation in AD models [82,83]. The broad-spectrum inhibition profile of VAS2870 could be advantageous in addressing the multifaceted nature of AD pathology. However, this non-specificity also raises concerns about potential off-target effects and the need for careful dosing to balance efficacy and safety. In summary, while DPI, apocynin, GKT137831, NOX2ds-tat, and VAS2870 provide different mechanisms and approaches to inhibiting NOX in AD models, each inhibitor faces distinct challenges related to specificity, efficacy, and safety. Future research should focus on developing selective inhibitors that can specifically target relevant NOX isoforms in AD pathology while minimizing off-target effects.

## 5. Conclusions

We reviewed the roles of NOX isoforms related to AD progression and described their respective contributions in the pathogenesis of AD. NOX enzymes, especially NOX2, contribute significantly to AD by increasing superoxide production in microglia and astrocytes, which drives neuronal death and perpetuates oxidative stress and neurodegeneration [84]. Similarly, NOX4 has been identified as a critical enzyme in AD pathology, particularly in the oxidative stress of astrocytes and its ferroptosis [25]. The expression of NOX1 and NOX3 is increased in AD brains [43]. Also, DUOX is elevated in aged brains and Drosophila models of AD [69,70]. Given that extensive research has not yet been conducted on NOX1, NOX3, DUOX1, and DUOX2, further investigation is warranted to elucidate the role of these NOX isoforms. While all NOX isoforms potentially contribute to AD pathogenesis, current evidence suggests that NOX2 and NOX4 contribute to AD pathogenesis and are the most promising targets for therapeutic intervention. The predominant role of NOX2 in microglial activation and neuroinflammation, coupled with the involvement of NOX4 in astrocytic oxidative stress and ferroptosis and in neuronal tauopathy, make these two isoforms particularly significant in AD progression [85,86]. Since the generation of ROS by NOX isoforms may be critical in the pathogenesis of AD, the understanding of NOX isoform-mediated underlying mechanisms during AD progression can help to develop a new approach for the treatment of AD. Future studies should focus on developing specific inhibitors for NOX2 and NOX4, while also exploring the potential contributions of other NOX isoforms to provide a comprehensive therapeutic strategy for AD.

## Figures and Tables

**Figure 1 ijms-25-12299-f001:**
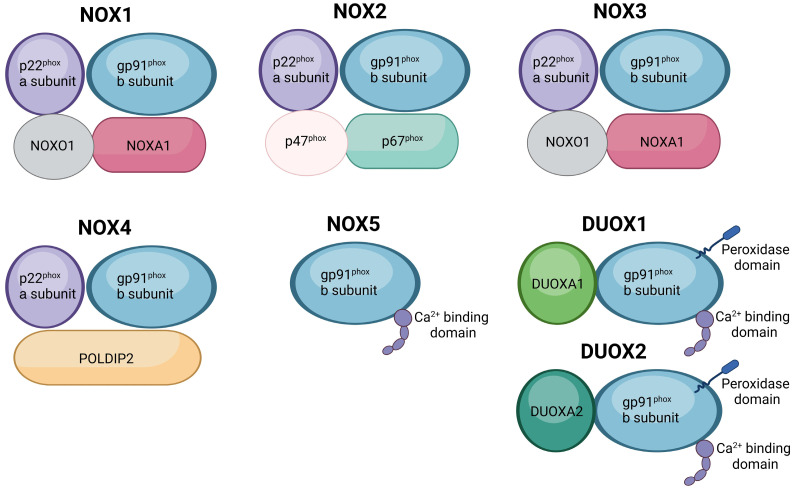
Summary for the structures of NOX isoforms. NOX1 NADPH oxidase 1, NOX2 NADPH oxidase 2, NOX4 NADPH oxidase 4, NOX5 NADPH oxidase 5, DUOX dual oxidases.

**Figure 2 ijms-25-12299-f002:**
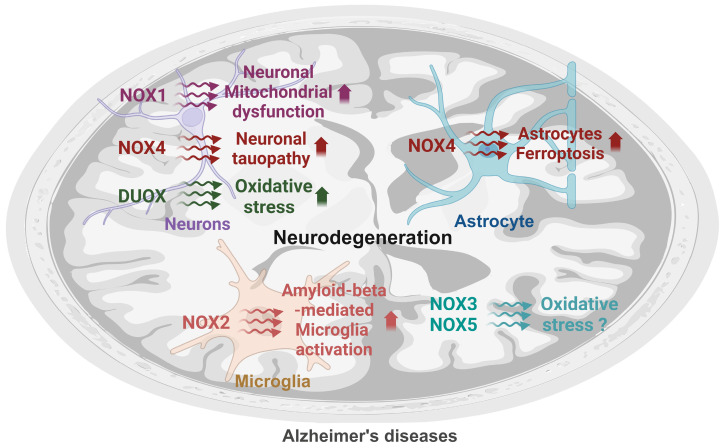
The summary for the role of NOX isoforms in Alzheimer’s disease. NOX1 NADPH oxidase 1, NOX2 NADPH oxidase 2, NOX4 NADPH oxidase 4, NOX5 NADPH oxidase 5, DUOX dual oxidases. Arrows mean upregulation.

**Table 1 ijms-25-12299-t001:** Structural features, activation mechanisms, and primary cell types of NOX isoforms.

NOX Isoform	Structural Features and Activation Mechanism	Cell Types
NOX1	Interacts with p22phox; requires NOXA1 and NOXO1 for activation	Colon epithelialium, vascular smooth muscle cells
NOX2	Interacts with p22phox; requires p67phox, p47phox, and p40phox for activation	Neutrophils, macrophages, microglia
NOX3	Interacts with p22phox; requires NOXA1 and NOXO1 for activation	Inner ear, cochlea cells
NOX4	Interacts with p22phox; constitutively active, no additional factors required	Fibroblasts, endothelial cells
NOX5	Forms homo- or multimeric structures; EF-calcium-binding domains; Ca^2+^-dependent activation	T-cells, vascular smooth muscle cells
DUOX1	Exhibits EF-calcium-binding domains and peroxidase domain; Ca^2+^-dependent, requires DUOXA1	Thyroid, epithelial cells
DUOX2	Exhibits EF-calcium-binding domains and peroxidase domain; Ca^2+^-dependent, requires DUOXA2	Thyroid cells, airway epithelium

**Table 2 ijms-25-12299-t002:** Summary of NOX isoform roles in Alzheimer’s disease.

NOX Isoform	Main Cellular Sources	Key Roles in AD	Associated Signaling Pathways
NOX1	Microglia	Involved in microglial activation and neuroinflammation	Regulation by Rac1 GTPase, linked with oxidative stress and inflammatory cytokine production
NOX2	Neurons, Microglia, Astrocytes	Aβ-stimulated ROS production, exacerbates synaptic loss and neuroinflammation	Activation by Aβ, involves p47phox and p67phox translocation; pathways linked to IL-1β, TNF-α, and IL-6 upregulation
NOX3	CNS Neurons	Contributes to oxidative stress and neuroinflammation	NOXO1 and p67phox regulation; less dependency on Rac GTPase
NOX4	Neurons, Astrocytes	Linked to ferroptosis and tau pathology	Constitutively active, hydrogen peroxide generation; linked to autophagic flux and lipid peroxidation
NOX5	Oligodendrocytes, Glioblastoma Cells	Implicated in BBB disruption and cognitive decline	Calcium-dependent activation, involves COX2 and TXA2S signaling for inflammation
DUOX1/2	Oligodendrocytes	Contributes to neuroinflammation and oxidative stress	Regulated by DUOXA1/2, associated with age-dependent neurodegeneration and lifespan regulation in models

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
