# Peer review of "Molecular Roles of NADPH Oxidase-Mediated Oxidative Stress in Alzheimer’s Disease: Isoform-Specific Contributions"

_ijms, 2024, doi:10.3390/ijms252212299_

Round 1
Reviewer 1 Report (New Reviewer)
Comments and Suggestions for Authors
The review titled " Molecular roles of NADPH oxidase-mediated oxidative stress in Alzheimer's diseases: Isoform-specific contributions" is an interesting work. It is well-organized and can serve as a valuable reference in this field.
Introduction
The introduction provides an excellent overview of the subject and effectively introduced the review parts.
Structure of NOX isoforms
You have to merge these sentences “Due to its structural distinctiveness, recent studies have increasingly focused on 94 NOX4 in AD pathogenesis. Due to its structural distinctiveness, recent studies have high- 95 lighted the roles of NOX4 in tau hyperphosphorylation and cognitive decline on AD mod- 96 els”
You have to rephrase these sentences “NOX5 does not engage with p22phox, instead forming homo- or multimeric struc- 98 tures [42]. Also, NOX5, DUOX1, and DUOX2 exhibit additional structural features includ- 99 ing EF-calcium-binding domains and the peroxidase domains present in DUOX enzymes 100 [43-46]. Unlike other NOX isoforms, NOX5, DUOX1, and DUOX2 do not require assis- 101 tance from other proteins for activation [47]”
It will be more representative to add a figure with the conservative structure of NOX
It will be more representative to add a table with the 7isoforms and structure particularity and cell types …..
The role of NOX isoforms
The figure 1 is not innovative I think it will be more representative to change it to a table
It would be advantageous to provide a more comprehensive regulation with the signalling pathways.
NOX inhibitors
You've just presented the results of previous studies without discussing them !!
It will be more important to discuss the results and explain more.
Many references can be removed
Throughout the article, you need to group the results and place them in a logical order according to the idea you want to convey
Author Response
Reviewer 1
The review titled " Molecular roles of NADPH oxidase-mediated oxidative stress in Alzheimer's diseases: Isoform-specific contributions" is an interesting work. It is well-organized and can serve as a valuable reference in this field.
Introduction
The introduction provides an excellent overview of the subject and effectively introduced the review parts.
- Structure of NOX isoforms
- You have to merge these sentences “Due to its structural distinctiveness, recent studies have increasingly focused on 94 NOX4 in AD pathogenesis. Due to its structural distinctiveness, recent studies have high- 95 lighted the roles of NOX4 in tau hyperphosphorylation and cognitive decline on AD mod- 96 els”
- You have to rephrase these sentences “NOX5 does not engage with p22phox, instead forming homo- or multimeric struc- 98 tures [42]. Also, NOX5, DUOX1, and DUOX2 exhibit additional structural features includ- 99 ing EF-calcium-binding domains and the peroxidase domains present in DUOX enzymes 100 [43-46]. Unlike other NOX isoforms, NOX5, DUOX1, and DUOX2 do not require assis- 101 tance from other proteins for activation [47]”
- It will be more representative to add a figure with the conservative structure of NOX
- It will be more representative to add a table with the 7isoforms and structure particularity and cell types …..
- The role of NOX isoforms
- The figure 1 is not innovative I think it will be more representative to change it to a table
- It would be advantageous to provide a more comprehensive regulation with the signalling pathways.
- NOX inhibitors
- You've just presented the results of previous studies without discussing them !!
- It will be more important to discuss the results and explain more.
- Many references can be removed
Throughout the article, you need to group the results and place them in a logical order according to the idea you want to convey
Dear Reviewer 1,
Thank you sincerely for taking the time to review our manuscript. We sincerely appreciate your insightful comments and suggestions, which have been very helpful in improving the quality of our paper. We completely agree with your suggestions and have made revisions based on your feedback.
- Structure of NOX isoforms
- According to the Reviewer’s suggestions, the two sentences have been merged into one.
: Due to its structural distinctiveness, recent studies have increasingly focused on NOX4 in AD pathogenesis, highlighting its roles in tau hyperphosphorylation and cognitive decline in AD models.
- According to the Reviewer’s suggestions, these sentences have been rephrased.
: NOX5 exhibits distinct structural characteristics, forming homo- or multimeric complexes rather than interacting with p22phox [42]. Additionally, NOX5, DUOX1, and DUOX2 possess unique structural features, including EF-calcium-binding domains and peroxidase domains, which are found in DUOX enzymes [43-46]. In contrast to other NOX isoforms, NOX5, DUOX1, and DUOX2 do not require assistance from other proteins for activation [47].
1.3 According to the Reviewer’s suggestions, we added a figure with the conservative structure of NOX in New Figure 1.
1.4 According to the Reviewer's suggestions, we have incorporated a table summarizing the structural features and cell types for the 7 NOX/DUOX isoforms. New Table 1 provides a clear overview of the key structural characteristics, activation mechanisms, and primary cell types expressing each isoform.
NOX isoform |
Structural features and activation mechanism |
Cell types |
NOX1 |
Interacts with p22phox ; Requires NOXA1 and NOXO1 for activation |
Colon epithelialium, vascular smooth muscle cells |
NOX2 |
Interacts with p22phox; Requires p67phox, p47phox, and p40phox for activation |
Neutrophils, macrophages, microglia |
NOX3 |
Interacts with p22phox; Requires NOXA1 and NOXO1 for activation. |
Inner ear, cochlea cells |
NOX4 |
Interacts with p22phox; Constitutively active, no additional factors required |
Fibroblasts, endothelial cells |
NOX5 |
Forms homo- or multimeric structures; EF-calcium-binding domains; Ca2+ dependent activation |
T-cells, vascular smooth muscle cells |
DUOX1 |
Exhibits EF-calcium-binding domains and peroxidase domain; Ca2+ dependent, requires DUOXA1 |
Thyroid, Epithelial cells |
DUOX2 |
Exhibits EF-calcium-binding domains and peroxidase domain; Ca2+ dependent, requires DUOXA2 |
Thyroid cells, airway epithelium |
Table 1. Structural features, activation mechanisms, and primary cell types of NOX isoforms.
- The role of NOX isoforms
: According to the Reviewer’s suggestions, we added the new Table 2 which includes main cellular sources, their roles in AD and additional section on comprehensive regulation with the associated signaling pathways.
NOX Isoform |
Main cellular sources |
Key roles in AD |
Associated signaling pathways |
NOX1 |
Microglia |
Involved in microglial activation and neuroinflammation |
Regulation by Rac1 GTPase, linked with oxidative stress and inflammatory cytokine production |
NOX2 |
Neurons, Microglia, Astrocytes |
Aβ-stimulated ROS production, exacerbates synaptic loss and neuroinflammation |
Activation by Aβ, involves p47phox and p67phox translocation; pathways linked to IL-1β, TNF-α, and IL-6 upregulation |
NOX3 |
CNS Neurons |
Contributes to oxidative stress and neuroinflammation |
NOXO1 and p67phox regulation; less dependency on Rac GTPase |
NOX4 |
Neurons, Astrocytes |
Linked to ferroptosis and tau pathology |
Constitutively active, hydrogen peroxide generation; linked to autophagic flux and lipid peroxidation |
NOX5 |
Oligodendrocytes, Glioblastoma cells |
Implicated in BBB disruption and cognitive decline |
Calcium-dependent activation, involves COX2 and TXA2S signaling for inflammation |
DUOX1/2 |
Oligodendrocytes |
Contributes to neuroinflammation and oxidative stress |
Regulated by DUOXA1/2, associated with age-dependent neurodegeneration and lifespan regulation in models |
Table 2. Summary of NOX Isoform roles in Alzheimer's Diseases.
- NOX inhibitors
According to the Reviewer’s suggestions, the following paragraph has been revised to include discussion and explanations on the significance and limitations of each study’s findings.
: Numerous studies have reported on the significant involvement of various NOX isoforms in AD pathogenesis. Therefore, understanding the inhibitors of these NOX isoforms could significantly contribute to new therapeutic approaches for AD. However, the complexity of NOX functions and the broad impacts of some inhibitors present both opportunities and challenges for their clinical applications. These inhibitors can be broadly categorized into non-specific and isoform-specific inhibitors, each with their own advantages and limitations. Among the various NOX inhibitors on AD, the earliest discovered and most well-known are diphe-nyleneiodonium (DPI) and apocynin [97]. DPI suppresses superoxide production by inhibiting flavoproteins, thereby affecting various enzymes beyond NOX, including nitric oxide synthase and xanthine oxidase [98]. While DPI has shown potential in reducing neuroinflammation and oxidative stress in AD models, its broad inhibitory effects on multiple flavin-dependent enzymes limit its clinical potential as a specific NOX inhibitor [99-101]. This lack of specificity poses a risk for off-target effects, as the inhibition of other flavoproteins could lead to unexpected side effects in a clinical setting. Therefore, although DPI has been valuable in experimental AD models, its broad activity makes it less suitable for human therapies targeting NOX in AD. Also, apocynin has demonstrated neuroprotective effects in various neurodegenerative diseases by reducing oxidative stress and inflammation through the interference of intracellular translocation in p47phox and p67phox subunits [102]. Unlike DPI, apocynin’s action appears more targeted within the NOX pathway, which may reduce broader systemic effects. Recent studies indicate that apocynin demonstrates efficacy in attenuating progression of AD by Aβ accumulation, oxidative stress, and neuroinflammation, potentially through modulation of BACE1 activity and transcription factors such as Nrf2 and NF-kB [103-105]. However, the exact mechanism of action of apocynin in AD is still not fully understood, and its efficacy may vary depending on the specific pathological context. Further research is needed to elucidate its long-term effects and optimal dosing strategies in AD patients. In addition to DPI and apocynin, other NOX inhibitors such as GKT137831, VAS2870, and Nox2ds-tat have attracted attention for their potential effects on AD pathology. GKT137831, described as a preferential direct inhibitor of NOX1 and NOX4, has shown excellent tolerability and reduced chronic inflammation markers in clinical trials [36,106]. This selective inhibition of NOX1 and NOX4 may offer a more targeted approach in managing AD pathology by reducing neuroinflammation while maintaining better tolerability. However, further trials are necessary to establish its efficacy and safety in long-term AD treatment. NOX2ds-tat, an 18-amino acid peptide and the first biological NOX inhibitor, has demonstrated significant efficacy in AD models [107]. When administered to aged Tg2576 mice or in studies using NOX2 knockout mice crossed with Tg2576 mice, NOX2ds-tat markedly reduced oxidative stress, enhanced neurovascular function, and alleviated behavioral impairments associated with the AD progression [108]. These findings suggest that NOX2ds-tat not only has the potential to reduce AD-related oxidative stress but also improves neurovascular health, a critical factor in AD pathology. VAS2870, identified by Vasopharm GmbH as a pan-NOX inhibitor, inhibits all NOX isoforms except NOX3 in cellular assays and particularly shows 40 % to 70 % inhibition of NOX2 and NOX4, meaning it is suitable for evaluation in AD models [37,109,110]. The broad-spectrum inhibition profile of VAS2870 could be advantageous in addressing the multifaceted nature of AD pathology. However, this non-specificity also raises concerns about potential off-target effects and the need for careful dosing to balance efficacy and safety. In summary, while DPI, apocynin, GKT137831, NOX2ds-tat, and VAS2870 provide different mechanisms and approaches to inhibiting NOX in AD models, each inhibitor faces distinct challenges related to specificity, efficacy, and safety. Future research should focus on developing selective inhibitors that can specifically target relevant NOX isoforms in AD pathology while minimizing off-target effects.
- Many references can be removed
: I sincerely appreciate your thoughtful feedback regarding the number of references in our review paper. We completely agree with your suggestion to remove some references. As you advised, we have reduced the total number of references from 117 to 94. We believe this revision will enhance the clarity and focus of our manuscript.
- Throughout the article, you need to group the results and place them in a logical order according to the idea you want to convey
: We fully agree with your comments, and appreciate the thoughtful feedback. In response to your recommendations, we have implemented the following changes:
- In the sections "Structure of NOX isoforms" and "The role of NOX isoforms," we have added comprehensive tables to provide readers with a clear overview of the entire content at a glance.
- For the "NOX inhibitors" section, we have restructured the content into two main categories: non-specific and isoform-specific inhibitors. This reorganization aims to enhance readers' understanding of the different types of inhibitors and their potential applications.
- We have carefully reordered the content to follow a more logical sequence. The manuscript now progresses from explaining the basic structure of NOX isoforms to detailing their specific roles in Alzheimer's disease, and finally discussing the potential of NOX inhibitors as therapeutic agents.
We believe these changes significantly improve the overall structure and readability of our manuscript, making it more accessible to our readers. Your feedback has been invaluable in helping us refine and enhance the quality of our work. Thank you once again for your valuable input. We hope that these revisions meet your expectations and contribute to a more comprehensive and logically structured review.
Reviewer 2 Report (New Reviewer)
Comments and Suggestions for Authors
General comments:
The review article entitled " Molecular roles of NADPH oxidase-mediated oxidative stress in Alzheimer's diseases: Isoform-specific contributions" by Junhyung Kim, Jong-Seok Moon, aimed to provide a review on the role of NADPH oxidase (NOX) enzymes in Alzheimer’s diseases and the possibility of the use of NOXs inhibitors as pharmacological agents to treat these diseases. In general this review is interesting but some points should be added.
Specific comments:
1)-The Introduction: line 53: “NADPH reduction”: should be “NADPH oxidation”.
2)-The Introduction: line 56: “O2.- contributes to the destruction of pathogens”: This is not true because O2.- itself is not very toxic. it should be “O2.- generates other H2O2 and other ROS that contribute.....”.
3)-The “Structure of NOX isoforms”: line 80: “cytochrome blight chain”, a space after “b” is missing, it should be “cytochrome b light chain”.
4)-The “Structure of NOX isoforms”: line 83: “p47phox/p40phox”, p40phox should be omitted because it is a different component not related to NOXO1.
5)-A Figure showing different NOXs should be added.
Author Response
Reviewer 2
General comments:
The review article entitled " Molecular roles of NADPH oxidase-mediated oxidative stress in Alzheimer's diseases: Isoform-specific contributions" by Junhyung Kim, Jong-Seok Moon, aimed to provide a review on the role of NADPH oxidase (NOX) enzymes in Alzheimer’s diseases and the possibility of the use of NOXs inhibitors as pharmacological agents to treat these diseases. In general this review is interesting but some points should be added.
Specific comments:
- The Introduction: line 53: “NADPH reduction”: should be “NADPH oxidation”.
- The Introduction: line 56: “O2.- contributes to the destruction of pathogens”: This is not true because O2.- itself is not very toxic. it should be “O2.- generates other H2O2 and other ROS that contribute.....”.
- The “Structure of NOX isoforms”: line 80: “cytochrome blight chain”, a space after “b” is missing, it should be “cytochrome b light chain”.
- The “Structure of NOX isoforms”: line 83: “p47phox/p40phox”, p40phox should be omitted because it is a different component not related to NOXO1.
- A Figure showing different NOXs should be added.
Dear Reviewer 2,
We sincerely appreciate your kind and insightful comments on our manuscript. We are grateful for your review and the valuable suggestions you have provided. Your detailed attention, especially in identifying specific textual corrections, has been invaluable for enhancing the accuracy and clarity of our work. We fully agree with your specific comments and are thankful for your constructive feedback, which has not only helped us refine our current work but also offered valuable insights for our future research endeavors. We believe that this revision significantly enriches our paper and provides readers with a more comprehensive perspective on the subject.
- According to the Reviewer's suggestions, the term "NADPH reduction" has been corrected to "NADPH oxidation" in the Introduction
: All NOX family enzymes are transmembrane proteins that traverse the membrane six times and produce O2•− from oxygen through NADPH oxidation, utilizing a heme-dependent mechanism [20].
- According to the Reviewer's suggestions, the following description has been changed in the introduction section
: In activation conditions, O2•− generates Hâ‚‚Oâ‚‚ and other ROS that contribute to the de-struction of pathogens [15].
- According to the Reviewer's suggestions, the following sentence has been corrected in the "Structure of NOX isoforms" section
: NOX1, NOX2, and NOX3 interact with the small transmembrane protein p22phox, also known as the human neutrophil cytochrome b light chain (CYBA) [32,33].
- According to the Reviewer's suggestions, the following sentence has been revised by removing the term p40phox.
: Activation of NOX1 and NOX3 requires interaction with cytosolic subunits NADPH oxi-dase activator 1 (NOXA1) and NADPH oxidase organizer 1 (NOXO1), which are homologous to p67phox and p47phox in NOX2, respectively [34,35].
- According to the Reviewer’s suggestions, we added the new Table 2 which includes main cellular sources, their roles in AD and additional section on comprehensive regulation with the associated signaling pathways.
Also, we added a figure with the conservative structure of NOX in New Figure 1.
NOX isoform |
Main cellular sources |
Key roles in AD |
Associated signaling pathways |
NOX1 |
Microglia |
Involved in microglial activation and neuroinflammation |
Regulation by Rac1 GTPase, linked with oxidative stress and inflammatory cytokine production |
NOX2 |
Neurons, Microglia, Astrocytes |
Aβ-stimulated ROS production, exacerbates synaptic loss and neuroinflammation |
Activation by Aβ, involves p47phox and p67phox translocation; pathways linked to IL-1β, TNF-α, and IL-6 upregulation |
NOX3 |
CNS Neurons |
Contributes to oxidative stress and neuroinflammation |
NOXO1 and p67phox regulation; less dependency on Rac GTPase |
NOX4 |
Neurons, Astrocytes |
Linked to ferroptosis and tau pathology |
Constitutively active, hydrogen peroxide generation; linked to autophagic flux and lipid peroxidation |
NOX5 |
Oligodendrocytes, Glioblastoma cells |
Implicated in BBB disruption and cognitive decline |
Calcium-dependent activation, involves COX2 and TXA2S signaling for inflammation |
DUOX1/2 |
Oligodendrocytes |
Contributes to neuroinflammation and oxidative stress |
Regulated by DUOXA1/2, associated with age-dependent neurodegeneration and lifespan regulation in models |
Table 2. Summary of NOX Isoform roles in Alzheimer's Diseases.
Once again, we appreciate your time and expertise in reviewing our work. Your feedback has been instrumental in improving our paper.
Round 2
Reviewer 1 Report (New Reviewer)
Comments and Suggestions for Authors
In this version of the review, “Molecular roles of NADPH oxidase-mediated oxidative stress in Alzheimer's diseases: Isoform-specific contributions.” We can see an acceptable evolution compared to the first version because it has become more structured with more explanation.
the authors have taken the reviewer's remarks and suggestions into consideration, which has positively impacted the quality and consistency of the review.
with this version, the review shows an excellent scientific level and represents an added value in the interested research topics
the review is accepted for me with this version
This manuscript is a resubmission of an earlier submission. The following is a list of the peer review reports and author responses from that submission.
Round 1
Reviewer 1 Report
Comments and Suggestions for Authors
In the submitted paper, the authors only listed the basic information about the NOX isoforms without attempting to correlate the relationship between their structures and involvement in Alzheimer's disease. Additionally, the authors mentioned that all five NOX isoforms could contribute to the pathogenesis of AD, leaving readers confused about which isoform is more significant based on their opinion. To improve this manuscript, the authors should also consider studies that focus on NOX as a potential target in AD, if such studies exist. Compare them and give readers their opinion.
Minor Remarks:
Only 20 out of 83 cited manuscripts (24%) are from the period 2020-2025. For a review paper, more recent publications should be included.
Author Response
Reviewer 1
In the submitted paper, the authors only listed the basic information about the NOX isoforms without attempting to correlate the relationship between their structures and involvement in Alzheimer's disease. Additionally, the authors mentioned that all five NOX isoforms could contribute to the pathogenesis of AD, leaving readers confused about which isoform is more significant based on their opinion. To improve this manuscript, the authors should also consider studies that focus on NOX as a potential target in AD, if such studies exist. Compare them and give readers their opinion.
According to the Reviewer’s suggestions, we revised key points including the relationship between the structures of NOX isoforms and their involvement in AD and the significant NOX isoforms related to AD following your comments. We appreciate your valuable comments. Additionally, we provided the description for the results of pharmacological and genetic NOX inhibition in AD.
In terms of the relationship between the structures of NOX isoforms and their involvement in AD, we revised and described more clearly.
To reflect new description, the following text has been added as below:
Abstract, page 1 line 18-20. “In this review, we provide an overview of all NOX isoforms in AD and their respective structural contribution to AD progression, highlighting NOX enzymes as a potential therapeutic target. Understanding the specific roles of respective NOX isoforms and their inhibitors could offer new insights into the understanding of AD pathology and the development of targeted treatments of AD.”
Introduction, page 2 line 67-69. “In this review, we provide an overview of all NOX isoforms in AD and their respective structural contribution to AD progression, highlighting NOX enzymes as a potential therapeutic target. Understanding the specific roles of respective NOX isoforms and their inhibitors could offer new insights into the understanding of AD pathology and the development of targeted treatments of AD.”
Section 2.1, page 2 line 77-89. “Activation of NOX1 and NOX3 requires interaction with cytosolic subunits NADPH oxidase activator 1 (NOXA1) and NADPH oxidase organizer 1 (NOXO1), which are homologous to p67phox and p47phox/p40phox in NOX2, respectively [34,35]. Structurally, the constitutive activation of NOX1 due to NOXO1's membrane localization and its dependence on NOXA1 for activation contributes to chronic oxidative stress and Aβ-induced neurotoxicity in AD brains [36-38]. The structural involvements of NOX2 isoform in AD pathology have also been elucidated through postmortem analyses of brain tissues and will be further discussed in the following sections. While NOX4 interacts with p22phox, it differs from other NOX enzymes as it functions without requiring cytosolic subunits for activation [39]. This structural autonomy of NOX4, combined with its ability to generate H2O2, allows for continuous low-level ROS production, making it a unique contributor to chronic oxidative stress in AD pathology [40,41]. Due to its structural distinctiveness, re-cent studies have increasingly focused on NOX4 in AD pathogenesis.”
Section 2.1, page 3 line 98-103. “The Ca2+-binding structure of NOX5, which is essential for its calcium-dependent activation, has been shown to impact blood-brain barrier integrity and memory loss in aging mice, indicating a potential role in AD progression [49,50]. The unique structural feature of an N-terminal peroxidase-like domain in DUOX1 and DUOX2 enables direct hydrogen peroxide generation, potentially contributing to the altered redox balance observed in brains of AD [51-53].”
Section 2.2.2, page 4 line 134-137. “With structural involvement, post-mortem analyses of AD brains reveal that NOX2 activation, indicated by the translocation of its subunits such as p47phox and p67phox to the cell membrane, significantly contributes to oxidative damage and neuroinflammation [56,63,66].”
In terms of the significant NOX isoforms related to AD, we revised the Conclusion section.
To reflect new description, the following text has been added as below:
Conclusions, page 6 line 248-249. “Similarly, NOX4 has been identified as a critical enzyme in AD pathology, particularly in oxidative stress of astrocytes and its ferroptosis [75].”
Conclusions, page 6 line 253-258. “While all NOX isoforms potentially contribute to AD pathogenesis, current evidence suggests that NOX2 and NOX4 contribute to AD pathogenesis and are the most promising targets for therapeutic intervention. The predominant role of NOX2 in microglial activation and neuroinflammation, coupled with the involvement of NOX4 in astrocytic oxidative stress and ferroptosis, make these two isoforms particularly significant in AD progression [110-115].”
Minor Remarks:
Only 20 out of 83 cited manuscripts (24%) are from the period 2020-2025. For a review paper, more recent publications should be included.
According to the Reviewer’s suggestions, we have significantly expanded our description and reference list to include more recent publications. The total number of references has increased from 83 to 115, with 41 of these being from 2020 onwards. Furthermore, we now have 50 references from the past five years, which substantially improves the currency of our description. The addition of recent literature has greatly enriched our paper, providing a more comprehensive and up-to-date overview of the field. We are grateful for your guidance, which has undoubtedly enhanced the quality and relevance of our manuscript. We sincerely appreciate your careful review and valuable feedback.
Reviewer 2 Report
Comments and Suggestions for Authors
This is a short review article that provides an update to the evolving story that NADPH oxidases (NOX's) account for the majority of the brain's oxidative stress (OS) burden and are potential targets for neurodegenerative diseases like Alzheimer's disease (AD). The authors build on a substantial literature accumulated over the last 10-15 years about the NOX family's (isoforms) involvement in many systemic diseases associated with OS damage. They present compelling evidence that certain NOX isoforms likely contribute to the microvascular and neuronal damage and pathological markers found in the AD brain.
I would have found it helpful for the authors to present a brief discussion about NOX inhibitors, where the field stands and what directions the field should move in. This is a critical section for any review article, whose purpose is both to educate and stimulate future research. I feel that this is a major deficiency of this paper.
Overall their paper is well written, accurate and easy to read. The figures included are helpful. I find no deficiencies in English spelling or grammar. I feel that their paper would substantially benefit from a discussion of NOX inhibitors. Clearly this is a popular and important area for future basic and clinical research
Author Response
Reviewer 2
This is a short review article that provides an update to the evolving story that NADPH oxidases (NOX's) account for the majority of the brain's oxidative stress (OS) burden and are potential targets for neurodegenerative diseases like Alzheimer's disease (AD). The authors build on a substantial literature accumulated over the last 10-15 years about the NOX family's (isoforms) involvement in many systemic diseases associated with OS damage. They present compelling evidence that certain NOX isoforms likely contribute to the microvascular and neuronal damage and pathological markers found in the AD brain.
I would have found it helpful for the authors to present a brief discussion about NOX inhibitors, where the field stands and what directions the field should move in. This is a critical section for any review article, whose purpose is both to educate and stimulate future research. I feel that this is a major deficiency of this paper.
Overall their paper is well written, accurate and easy to read. The figures included are helpful. I find no deficiencies in English spelling or grammar. I feel that their paper would substantially benefit from a discussion of NOX inhibitors. Clearly this is a popular and important area for future basic and clinical research
We agree with your suggestion regarding the importance of NOX inhibitors in AD in our manuscript. We appreciate your valuable comments. According to the Reviewer’s suggestions, we have added a section that addresses the current state of NOX inhibitors, their potential as therapeutic agents, and the future directions of research in this area.
To reflect new description, the following text has been added as below:
Abstract, page 1 line 18-20. “In this review, we provide an overview of all NOX isoforms in AD and their respective structural contribution to AD progression, highlighting NOX enzymes as a potential therapeutic target. Understanding the specific roles of respective NOX isoforms and their inhibitors could offer new insights into the understanding of AD pathology and the development of targeted treatments of AD.”
Introduction, page 2 line 67-69. “In this review, we provide an overview of all NOX isoforms in AD and their respective structural contribution to AD progression, highlighting NOX enzymes as a potential therapeutic target. Understanding the specific roles of respective NOX isoforms and their inhibitors could offer new insights into the understanding of AD pathology and the development of targeted treatments of AD.”
Section 3, page 5-6 line 210-242. “Numerous studies have reported on the significant involvement of various NOX isoforms in AD pathogenesis. Therefore, understanding the inhibitors of these NOX isoforms could significantly contribute to new therapeutic approaches for AD. Among the various NOX inhibitors on AD, the earliest discovered and most well-known are diphenyleneiodonium (DPI) and apocynin [95]. DPI suppresses superoxide production by inhibiting flavoproteins, thereby affecting various enzymes beyond NOX, including nitric oxide synthase and xanthine oxidase [96]. While DPI has shown potential in reducing neuroinflammation and oxidative stress in AD models, its broad inhibitory effects on multiple flavin-dependent enzymes limit its clinical potential as a specific NOX inhibitor [97-99]. Also, apocynin has demonstrated neuroprotective effects in various neurodegenerative diseases by reducing oxidative stress and inflammation through the interference of intracellular translocation in p47phox and p67phox subunits [100]. Recent studies indicate that apocynin demonstrates efficacy in attenuating progression of AD by diminishing Aβ accumulation, oxidative stress, and neuroinflammation, potentially through modulation of BACE1 activity and transcription factors such as Nrf2 and NF-kB [101-103]. In addition to DPI and apocynin, other NOX inhibitors such as GKT137831, VAS2870, and Nox2ds-tat have attracted attention for their potential effects on AD pathology. GKT137831, described as a preferential direct inhibitor of NOX1 and NOX4, has shown excellent tolerability and reduced chronic inflammation markers in clinical trials [36,104]. NOX2ds-tat, an 18-amino acid peptide and the first biological NOX inhibitor, has demonstrated significant efficacy in AD models [105]. When administered to aged Tg2576 mice or in studies using NOX2 knockout mice crossed with Tg2576 mice, NOX2ds-tat markedly reduced oxidative stress, enhanced neurovascular function, and alleviated behavioral impairments associated with the AD progression [106]. VAS2870, identified by Vaso-pharm GmbH as a pan-NOX inhibitor, inhibits all NOX isoforms except NOX3 in cellular assays and particularly shows 40% to 70% inhibition of NOX2 and NOX4, meaning it is suitable for evaluation in AD models [37,107,108]. By targeting these enzymes, it may be possible to mitigate oxidative stress, neuroinflammation, and other pathological processes associated with AD. However, its therapeutic potential faces significant hurdles due to dose-dependent variability and functional specificity towards different NOX isoforms. Therefore, further research into NOX isoforms and selective NOX inhibitors could provide valuable insights and potentially lead to breakthrough treatments for AD pathology.”
Conclusions, page 6 line 261-263. “Future studies should focus on developing specific inhibitors for NOX2 and NOX4, while also exploring the potential contributions of other NOX isoforms to provide a comprehensive therapeutic strategy for AD.”
Round 2
Reviewer 1 Report
Comments and Suggestions for Authors
Although the manuscript has been improved over the previous version, the corrections made are not sufficient to accept this manuscript.